# Mechanisms and therapeutic implications of RTA 408, an activator of Nrf2, in subarachnoid hemorrhage–induced delayed cerebral vasospasm and secondary brain injury

**Tai-Hsin Tsai[1,2,3], Szu-Huai Lin[4], Chieh-Hsin Wu[1,2,3], Yi-Cheng Tsai[1,3], Sheau-Fang Yang[5,6], Chih-Lung Lin[1,2,3]***

1 Division of Neurosurgery, Department of Surgery, Kaohsiung Medical University Hospital, Kaohsiung, Taiwan, 2 Department of Surgery, School of Medicine, College of Medicine, Kaohsiung Medical University, Kaohsiung, Taiwan, 3 Graduate Institute of Medicine, College of Medicine, Kaohsiung Medical University, Kaohsiung, Taiwan, 4 Department of Nursing, Kaohsiung Medical University Hospital, Kaohsiung, Taiwan, 5 Department of Pathology, Kaohsiung Medical University Hospital, Kaohsiung, Taiwan, 6 Department of Pathology, School of Medicine, College of Medicine, Kaohsiung Medical University, Kaohsiung, Taiwan

* chihlung1@yahoo.com

**Data Availability Statement:** All relevant data are within the manuscript and its Supporting Information files.

## Abstract

### Objectives

More and more evidence suggests oxidative stress and inflammation contribute importantly to subarachnoid hemorrhage (SAH)-induced cerebral vasospasm and secondary brain injury. Recent evidence indicates Nuclear factor (erythroid-derived 2)-like 2 (Nrf2) increases the expression of antioxidant genes and decreases the expression of pro-inflammatory genes. This study examines the effects of an activator of Nfr2, RTA 408, on SAH-induced cerebral vasospasm and possible mechanism underlying its effect in a two-hemorrhage rodent model of SAH.

### Methods

We randomly assigned 60 Sprague-Dawley male rats (350 to 420g) to five groups twelve rats each: one control group (no SAH), one untreated SAH only group and three RTA-408 treatment groups (SAH+ RTA 408 0.5 mg/kg/day, SAH+RTA 408 1 mg/kg/day and a SAH +RTA 408 1.5 mg/kg/day). The treatment groups were administered RTA 408 by intraperitoneal injection thirty min following first induction of SAH for seven days starting with first hemorrhage. Cerebral vasospasm was determined by averaging the cross-sectional areas of basilar artery 7 days after first SAH. Expressions of Nrf2, NF-κB and iNOS in basilar artery and expressions of Nrf2, HO-1, NQO1 and Cleaved caspase-3 were evaluated. Tissue TNF-alpha was assessed by ELISA using the protein sampled from the dentate gyrus, cerebral cortex, and hippocampus.

**Funding:** This work was supported by the Ministry of Science and Technology, Taiwan, under Grant MOS 106-2314-B-037-025-MY2.

**Competing interests:** The authors have declared that no competing interests exist.

**Abbreviations:** AIMs, Antioxidant inflammation modulators; ARE, Antioxidant response element; CV, Cerebral vasospasm; DCI, Delayed cerebral ischemia; EBI, Early brain injury; HIF-1, Hypoxia-inducible factor 1; HO-1, heme oxygenase-1; iNOS, Inducible nitric oxide synthase; Keap1, Kelch-like ECH-associated protein 1; NFAT, Nuclear factor of activated T-cells; NF-κB, Nuclear factor kappa B; NO, Nitric oxide; NQO-1, NADPH quinone oxidoreductase-1; Nrf2, Nuclear factor (erythroid-derived 2)-like 2; ROS, Reactive oxygen species; SAH, Subarachnoid hemorrhage.

## Results

Prior to perfusion fixation, there were no significant physiological differences among the control and treated groups. RTA 408 treatment attenuated the morphological changes caused by cerebral vasospasm. It mitigated SAH-induced suppression of Nrf2 and increased expression of NF-κB and iNOS in the basilar artery. In dentate gyrus, it reversed SAH-decreases in Nrf2, HO-1, NQO-1 and cleaved caspase-3 and RTA 408 1.5 mg/kg/day reversed SAH increases in TNF-alpha.

## Conclusion

It was concluded that RTA 408 reversal vasospasm was achieved via increases in Nrf2 and decreases in NF-κB and iNOS. It exerted a neuron-protection effect by decreasing the apoptosis-related protein cleaved caspase-3 and decreasing the information cytokine TNF-alpha expression, which it achieved by increasing HO-1 and NQO-1 protein found downstream from Nrf2 and Nrf2. We believe that RTA 408 can potentially be used to manage of cerebral vasospasm and secondary brain injury following SAH.

## Introduction

Although cerebral vasospasm (CV)-induced delayed cerebral ischemia after aneurysmal subarachnoid hemorrhage (SAH) has been recognized for more than 50 years, its pathophysiology remains unknown, limiting the range of effective treatment strategies [1] Despite advances in the diagnosis and treatment of aneurysmal SAH, CV-induced delayed cerebral ischemia continues to be the leading but potentially treatable cause of death and disability in patients with ruptured cerebral aneurysms [2]. Many pathological processes have been proposed as possible mechanisms underlying delayed CV after SAH, including endothelial damage, smooth muscle contraction, changes in vascular responsiveness, and the inflammatory and/or immunological response of the vascular wall [3]. Apoptosis in the vasculature is reported to play a significant role in SAH, and apoptotic cascades may be responsible for vasospasm [4, 5] Therefore, the occurrence of apoptosis after SAH has important implications for both vasospasm and the long-term sequelae of SAH [6]. Furthermore, there is increasing evidence that oxidative stress and inflammation may be the leading causes of subarachnoid hemorrhage (SAH)–induced cerebral vasospasm and secondary brain injury [7].

The transcription factor Nuclear factor erythroid 2-related factor 2 (Nrf2) is encoded by the NFE2L2 gene in humans [8, 9]. Nrf2, a basic leucine zipper protein, protects against inflammation and oxidative damage triggered by injury through its regulation of the expression of antioxidant proteins [10]. Antioxidant inflammation modulators (AIMs), including synthetic derivatives of oleanolic acid, have been found to achieve their anti-inflammatory and anticarcinogenic effects by their activation of Nrf2 and inhibition of nuclear factor kappa B (NF-κB) activity [10]. AIMs activate Nrf2 by binding to the adaptor protein Kelch-like ECH-associated protein 1 (Keap1) and blocking its ability to promote Nrf2 degradation [11, 12]. However, when under inflammatory and oxidative stress, activated Nrf2 detaches from Keap1 and moves into the nucleus, there binding to the antioxidant response element [13], a 51-bp DNA element often observed in the promoter region of various genes encoding detoxification enzymes and cytoprotective proteins [14]. Once the Nrf2–ARE signaling pathway is activated, it regulates the expression of antioxidant and anti-inflammatory genes, which translates to

cytokines that play a role in antioxidant and anti-inflammatory defense in cells. Newly synthe-sized Nrf2 collects in the cell nucleus where it exerts antioxidant and anti-inflammatory effects by up-regulating antioxidant genes and down-regulating pro-inflammatory genes [15, 16] Nrf2 activation also confers potent anti-inflammatory properties on cells by detoxifying reac-tive oxygen species (ROS), resulting in the activation of pro-inflammatory transcription factor NF-κB. Several Nrf2 pathway-stimulating drugs are being investigated as potentially useful in the management of different diseases, including cancer, mitochondrial myopathies, metabolic and neurodegenerative disorders, as well as disorders involving inflammation and oxidative stress [17–19].

One recent animal study observed a new second generation semisynthetic oleanane triter-penoid, RTA 408, to activate Nrf2 in rat skin [20, 21]. Several lines of evidence have suggested that RTA 408 possesses antioxidative and anti-inflammatory activities [20, 21] and it has been found able to improve mitochondrial respiration and bioenergetics [22]. The mechanism underlying RTA 408 effects involves combined activation of the antioxidative transcription factor Nrf2 and the inhibition of the pro-inflammatory transcription factor NF-κB, creating an antioxidant and anti-inflammatory phenotype [20].

The role of RTA 408 in the treatment of subarachnoid hemorrhage–induced vasospasm and secondary brain injury has never been studied. The anti-inflammatory and antioxidant effects of RTA 408 constitute a potential therapeutic strategy in the treatment of oxidative stress and inflammation after SAH. Thus, this study examined the effect that RTA 408 might have on experimentally-induced SAH in rats.

## Materials and methods

### Animal treatment

All procedures were approved by the Kaohsiung Medical University Animal Care and Use Committee. The animals were purchased from BioLASCO Taiwan Co., Ltd. and individually housed in cages with unlimited access to food and water. Cages were place in a temperature (25°C) and humidity (40–50%)-controlled room with a 12 h light/dark cycle. We randomly assigned 60 Sprague-Dawley male rats (350 to 420g) in one of five groups twelve rats each: a control group (no SAH); 2), an untreated SAH only group and one of three RTA 408 treatment groups SAH+RTA 408 0.5 mg/kg/day, 1 mg/kg/day and 1.5 mg/kg/day. The treatment groups were administered RTA 408 by intraperitoneal injection starting 30 min after the first induc-tion of SAH for 7 days following the first hemorrhage. The RTA-408 was purchased from MedChemExpress (MCE), USA; Cat. No.: HY-12212.

### Induction of experimental SAH

A two-hemorrhage rodent model of SAH was adopted in this study. Briefly, under anesthesia and analgesia to minimize pain, the animals were anesthetized by an intraperitoneal injection of 40 mg/kg Zoletil 50® containing a mixture of zolazepam and tiletamine hypochloride (Vir-bac, Carros, France). The rectal temperature was controlled at 36±1°C with a heating pad (Harvard Apparatus). The tail artery was cannulated with a polyethylene catheter for monitor-ing blood pressure and heart rate. The cisterna magna was punctured percutaneously with a 25-gauge butterfly needle. About 0.10 to 0.15 ml of cerebrospinal fluid was slowly withdrawn and the junction between the needle and tube was clamped. Fresh autologous, nonheparinized blood (0.3 ml) was withdrawn from the tail artery. Using a needle-in-needle method (inserting a 30-gauge needle into the 25-gauge butterfly needle at the junction of needle and tube), we slowly injected blood into the cisterna magna. The same procedure was repeated 48 hours later. We kept the animals warm using heating pads to avoid burns and thermal bedding

during post-surgical recovery. Animals were checked frequently, approximately every 10–15 minutes, and turned side to side until they recovered. Seven days after the first SAH, the animals were deeply anaesthetized before they were euthanized by perfusion and fixation. All rats were humanely euthanized with intraperitoneal injection of Zoletil 50®. The brains were subsequently removed, placed in a fixative solution, and stored at 4°C overnight. We obtained samples of the basilar and brain after intra-cardiac perfusion with normal saline to measure protein contents of Nrf2 and Nrf2 related protein.

### Basilar artery morphometric analysis

Tissue morphometric analysis was performed in samples obtained in six animals from each group. The middle third of basilar artery was dissected and cut into thick cross sections (10 μm). They were then placed on glass slides and stained using hematoxylin and cosin for morphometric analysis. We randomly selected four arterial cross-sections from each animal to analyze. These cross-sections were analyzed by computer-assisted morphometry (ImageJ, NIH). ANOVA and the Bonferroni post-hoc test were used to compare groups. A $p < 0.05$ was considered significant.

### Western blot analysis of basilar artery Nrf2, NF-κB and iNOS and brain tissue Nrf2, heme oxygenase-1 (HO-1), NAD(P)H: Quinone oxidoreductase (NQO1), and cleaved caspase-3

We randomly selected six rats from each group and collected tissue samples from the basilar artery and brain, including the cortex, hippocampus and dentate gyrus. All samples were homogenized in ice-cold M-PER Mammalian Protein Extraction Reagent (Thermo) along with a protease inhibitor (Complete Mini; Roche) and centrifuged 15,000 rpm for 20 min. We measured protein levels by Bio-Rad protein microassay. Briefly, samples were heated in water at boiling for 5 min. Protein samples in equal amounts were loaded onto a polyvinylidene difluoride (PVDF; PerkinElmer) membrane electroblotting for 90 min (100V). The membrane was placed a Tween-Tris buffer saline solution (t-TBS; 20mM Tris base, 0.44mM NaCl, 0.1% Tween 20, pH 7.6) containing 5% nonfat dry milk and 0.1% Tween 20 overnight at 4°C. The next day the blots were incubated with following primary antibodies at the following dilutions: antibodies against rabbit polyclonal Nrf2 (Proteintech) 1:000, mouse monoclonal NF-κB (Cellsignaling) 1:500, mouse monoclonal iNOS (BD) 1:500, mouse monoclonal HO-1 (Ezno) 1:500, rabbit polyclonal NQO1 (Abcam) 1:1000, rabbit polyclonal cleaved caspase-3 (Cellsignaling) 1:500, and beta-actin (Sigma) 1:20000. They were then rinsed with t-TBS for 30 min and incubated with goat anti-mouse or anti-rabbit IgG antibody conjugated to horseradish peroxidase, followed by rinsing with t-TBS for 30 min and incubation with electrochemiluminescence reagent (PerkinElmer) for 2 min, according to manufacturer's directions.

### Measurement of Tumor Neurosis Factor-alpha (TNF-α)

ELISA was performed to assess tissue TNF-α using protein sampled from the cerebral cortex, hippocampus and dentate gyrus of six randomly selected rats in each group. TNF-α was quantified using a commercial kit (Invitrogen) according to manufacturer's directions. Spectrophotometry was used to measure TNF-α, following guidelines for performing the assay. Absorbance was measured at 450 nm.

## Statistics

Characteristic data were expressed as mean ± standard error. Images were tested using image assay software: ImageJ, NIH. Experimental group differences were tested using one-way analysis of variance and the Bonferroni post-hoc test by SPSS soft ware. A $p < 0.05$ was considered significant.

## Results

### General observations

Prior to perfusion, there were no significant differences between the control and treatment groups in body weight, mean arterial blood pressure or heart rate.

### RTA 408 reduced SAH-induced basilar artery vasospasm

SAH rats were found to have significantly reduced areas in cross-sections taken from basilar arteries in SAH rats compared to controls (24585.34±4361.925 μm$^2$ vs. (53478.24 ±5563.646 μm2) (54% reduction; $p < 0.01$) In SAH rats treated with RTA 408 (0.5 mg/kg/day) there was a 41% reduction, compared to controls (31532.49±5716.087 μm$^2$) ($p < 0.05$) (Fig 1). In SAH rats treated with RTA 408 1 mg/kg/day and 1.5 mg/kg/day the areas were measured to be 42256.36±4960.503 μm$^2$ and 50198.01±5835.611 μm$^2$, respectively, significantly different from the untreated SAH rats ($p < 0.05$ and $p < 0.01$, respectively). We found no significant difference between rats treated with 1.5 mg/kg/day of RTA 408 and the controls, suggesting complete reduction.

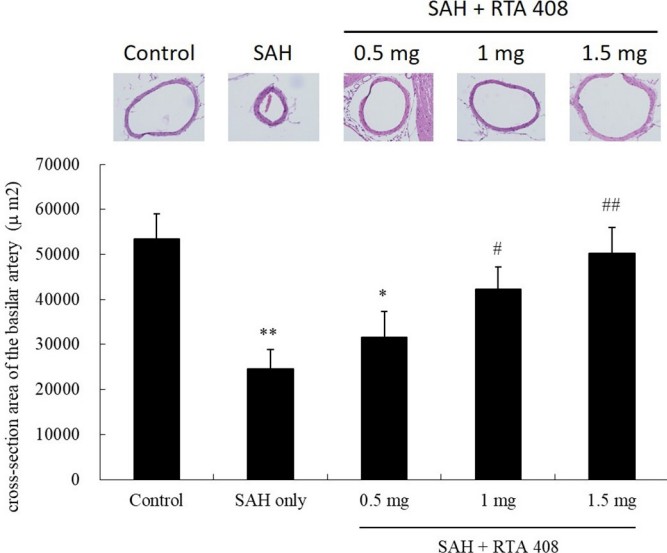

**Fig 1. The anti-spastic effect of RTA 408 of vasospasm of the basilar artery induced by SAH.** Compared with the control group, the areas in the SAH only were reduced by 54% ($p < 0.01$). The cross-sectional areas in the RTA 408 (1 mg/kg/day) treatment group and RTA 408 (1.5 mg/kg/day) treatment group differed significantly from those of the SAH-only group ($p < 0.05$, and $p < 0.01$, respectively). There was no significant difference between the RTA 408 (0.5 mg /kg/day) treatment group and the control group. $^*p < 0.05$ and $^{**}p < 0.01$ compared to controls. #$p < 0.05$, and ##$p$ $< 0.01$ compared to SAH only.

## RTA 408 up-regulated Nrf2 and down-regulated NF-κB and iNOS in the basilar artery

The SAH group had significantly lower basilar artery levels of Nrf2 protein than the controls ($p$ <0.01), while groups treated with RTA 408 1 and 1.5 mg/kg/day had significantly higher levels of Nrf2 protein than the SAH only group ($p$ <0.05 and $p$ <0.01, respectively). Those treated with RTA 408 0.5, 1, and 1.5 mg/kg/day did not have significantly different Nrf2 protein levels than the controls ($p$ >0.05) (Fig 2A and 2B). All groups treated with RTA 408 had higher levels of Nrf2 protein in the basilar artery than the controls ($p$ <0.01) (Fig 2).

The untreated SAH only rats and the rats treated with RTA 408 0.5 mg/kg/day were both found to have higher levels basilar artery NF-κB than the controls (both, $p$ <0.01). Compared with the untreated SAH only group, all RTA 408 treatment groups had significantly lower basilar artery NF-κB protein levels (RTA 408 0.5 mg/kg/day and 1 mg/kg/day both $p$ <0.05; 1.5 mg/kg/day $p$ <0.01). There was no significant difference found NF-κB protein concentration between either those treated with RTA 408 1 mg/kg/day or 1.5 mg/kg/day and the controls (Fig 2B).

Compared with controls, iNOS protein was increased significantly in the SAH only group and RTA 408 0.5 mg/kg/day group (both, $p$ <0.01) as well as the RTA 408 1.0 mg/kg/day group ($p$ <0.05). Compared with the SAH only group, SAH rats treated with RTA 408 1.5 mg/kg/day had significantly lower levels of basilar artery iNOS protein ($p$ <0.01), but their NF-κB protein levels were not significantly different from the controls (Fig 2B).

## RTA 408 exerted its neuroprotective effect by increasing Nrf2 and downstream HO-1 and NQO1

**Brain tissue Nrf2.**   We measured the expression of Nrf2 protein in cortex, hippocampus, and dentate gyrus after SAH in rats treated and untreated with RTA 408 (Fig 3). There was no difference in cortex or hippocampus Nrf2 between either the controls or SAH groups treated or untreated with RTA 408 treatment seven days post first SAH (Fig 3A). However, there was a significant decrease in dentate gyrus Nrf2 in the SAH only group, compared to controls ($p$ <0.05). Treatment with RTA 408 (0.5, 1, 1.5 mg/kg/day) significantly prevented the SAH induced reduction of Nrf2 in the dentate gyrus compared to the SAH only group ($p$ <0.05 in RTA 408 0.5 mg/kg/day and 1 mg/kg/day; $p$ <0.01 RTA 408 1.5 mg/kg/day). Compared to controls, however, there was no significant difference in Nrf2 levels in either those treated with RTA 408 0.5 mg/kg/day or 1 mg/kg/day. In the SAH group treated with RTA 408 1.5 mg/kg/day group, however, there was a significant increase in dentate gyrus Nrf2, compared to controls ($p$ <0.05) (Fig 3B).

**Brain tissue HO-1.**   We measured protein levels of HO-1 in the cortex, hippocampus, and dentate gyrus (Fig 3A). Compared to controls, HO-1 was significantly decreased in dentate gyrus ($p$ <0.05 but not in the cortex or hippocampus in SAH rats untreated with RTA 408. HO-1 was significantly increased in the hippocampus and dentate gyrus of SAH rats treated with either RTA 408 0.5, 1 or 1.5mg/kg/day, compared to the untreated SAH only group (both $p$ <0.05. However, we found no significant change in HO-1 in the cortex between SAH rats untreated and treated with RTA 408 treatment, compared to controls (Fig 3B).

**Brain tissue NQO1.**   We also measured protein levels of NQO1 in cortex, hippocampus, and dentate gyrus (Fig 3A). Compared to controls, we found it decreased in the dentate gyrus of the SAH group ($p$ <0.05). Compared to the SAH only group, those treated with RTA 408 1 and 1.5 mg/kg/day had significantly increased NQO1 levels in the dentate gyrus (p<0.05).

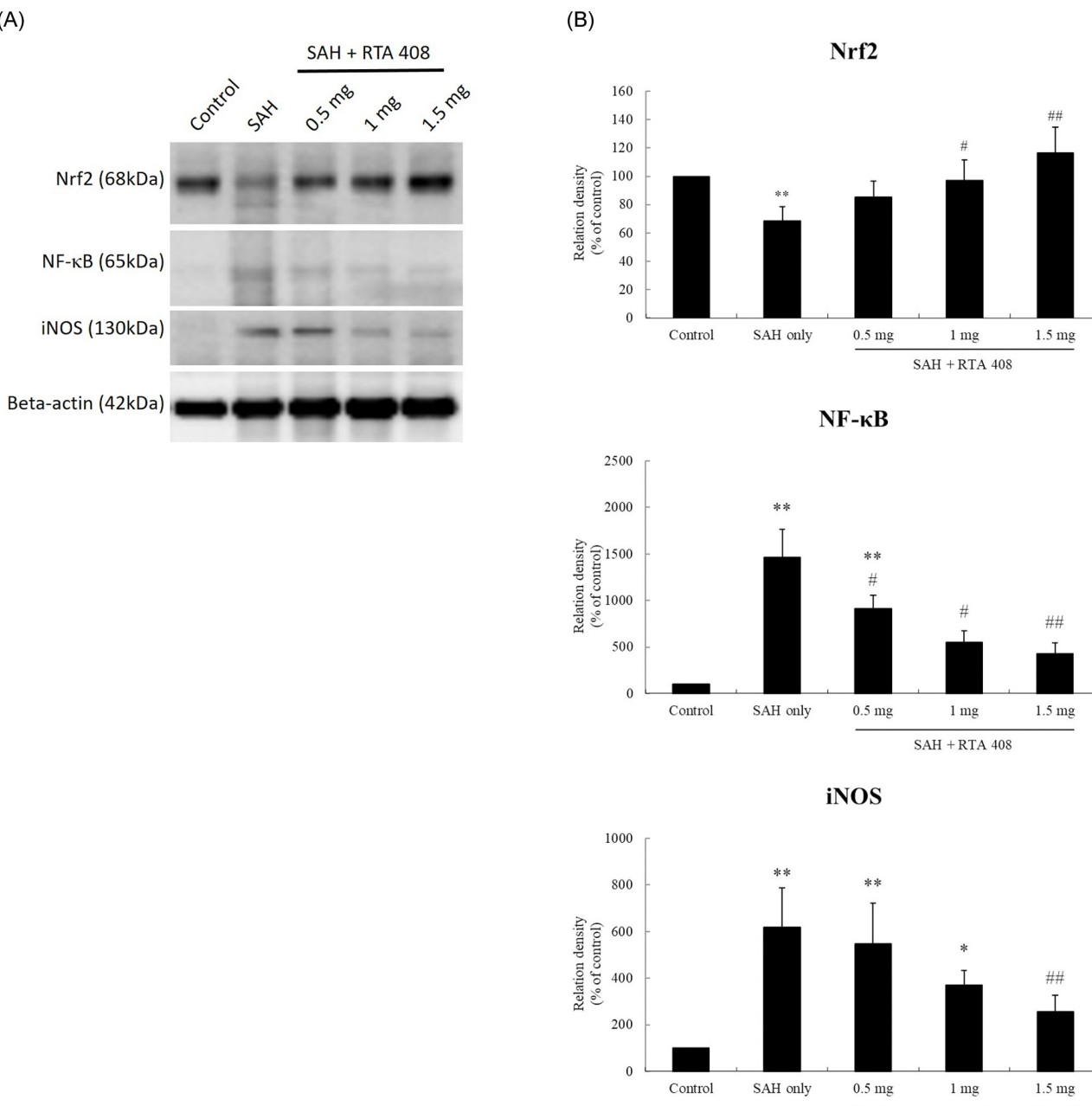

**Fig 2. The expression of Nrf2, NF-κB and iNOS in basilar artery after SAH and after RTA treatment.** (A) Western blots of Nrf2, NF-κB and iNOS in the basilar artery. (B) SAH rats had significant reductions in Nrf2, NF-κB and iNOS. RTA 408(1mg/kg/day and 1.5 mg/kg/day) treatment reversed these changes, up-regulating Nrf2 and down-regulating NF-κB and iNOS. $^*p <0.05$ and $^{**}p <0.01$ compared to controls. $\#p <0.05$, and $\#\#p <0.01$ compared to SAH only.

None of the RTA 408-treated groups had significantly different dentate gyrus levels of NQO1 from the controls. In addition, we found no significant change in the expression of NQO1 in the cortex and hippocampus between controls and untreated SAH rats or SAH rats treated with RTA 408 (Fig 3B).

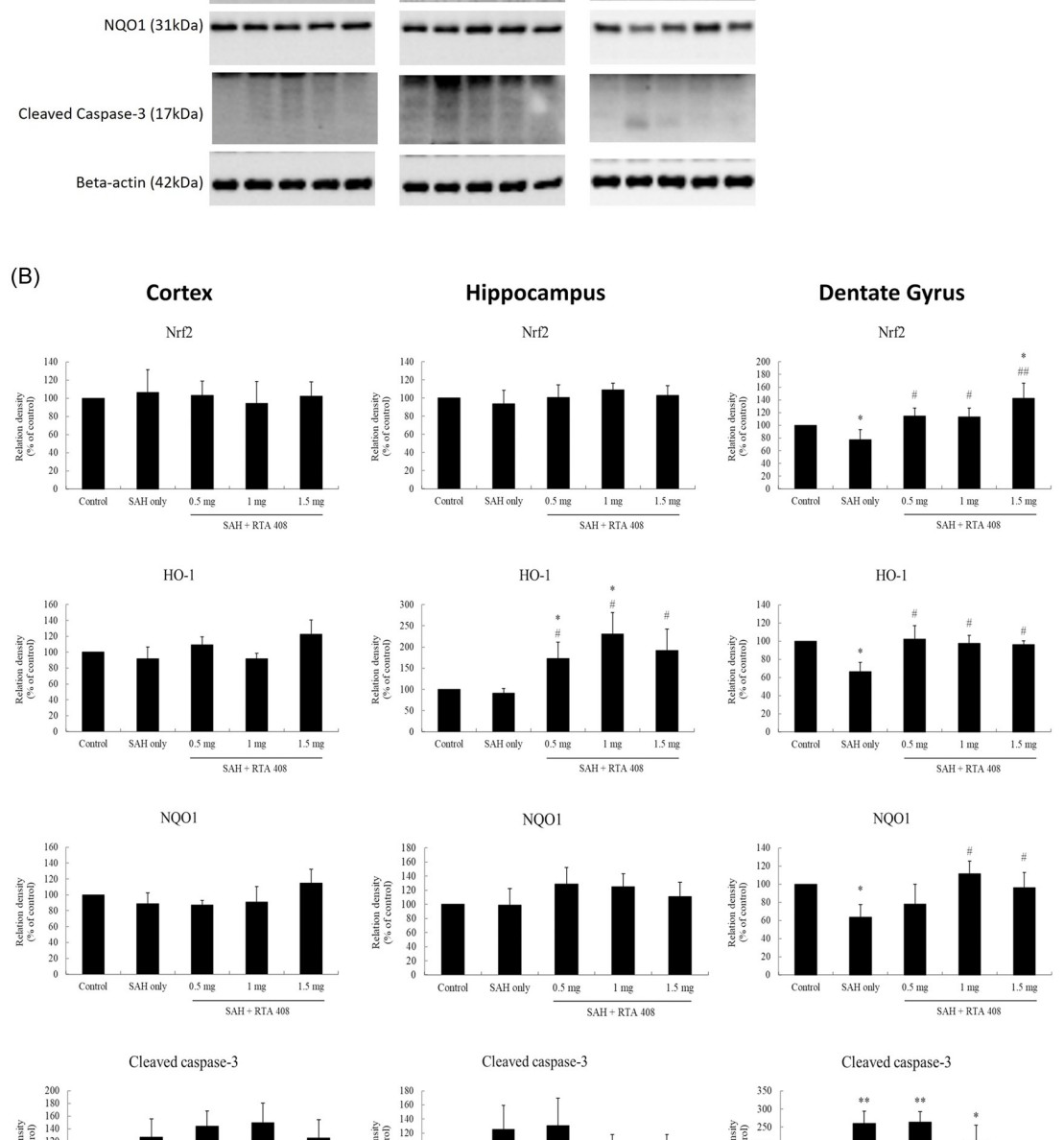

**Fig 3. Brains tissue expression of Nrf2, HO-1, NQO1 and cleaved caspase-3 protein.** (A) Western blots of Nrf2, HO-1, NQO1, cleaved caspase-3 and beta-action in the cortex, hippocampus, and dentate of rat brain. (B) Nrf2, HO-1 and NQO1 proteins were significantly reduced and cleaved caspase-3 was and the protein content of cleaved caspase-3 was significantly increased in the dentate gyrus untreated SAH rats. RTA-408 significantly increased Nrf2, HO-1 and NQO1 proteins and significantly decreased cleaved caspase 3 in the dentate gyrus of SAH rats.. There was no significant difference in these protein expressions in the cortexes of controls and untreated SAH rats and rats treated with RTA-408. $^*p < 0.05$ and $^{**}p < 0.01$ compared to controls. $\#p < 0.05$, and $\#\#p < 0.01$ compared to SAH only.

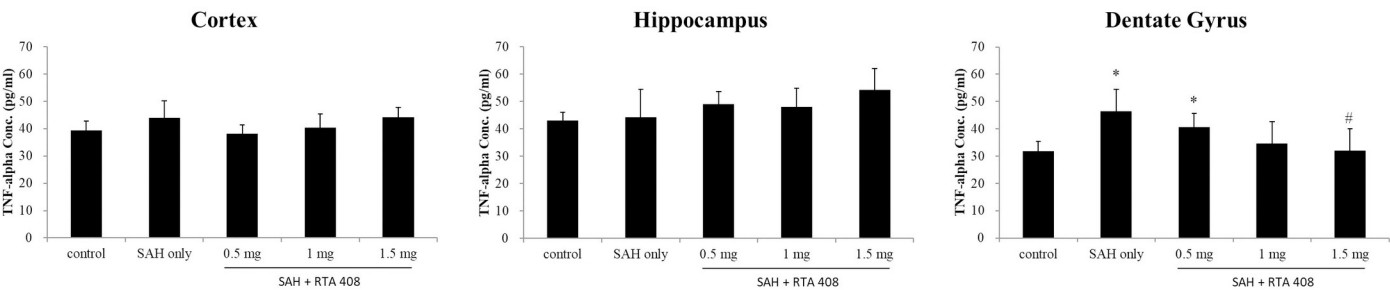

**Fig 4. ELISA assay for TNF-alpha.** TNF-alpha was found to be significantly increased in the dentate gyrus in untreated SAH rats. RTA 408 (1.5 mg/ml/kg) significantly reduced the TNF-alpha in the dentate gyrus in SAH rats. No significant difference TNF-alpha expression was noted in the cortex or hippocampus in control and SAH animals. $^{*}p <0.05$ and $^{**}p <0.01$ compared to controls. #$p <0.05$, and ##$p <0.01$ compared to SAH only.

## Anti-apoptotic effect of RTA 408 in dentate gyrus brought about by decreases in the cleaved caspase-3 and TNF-α

Apoptotic protein, cleaved caspase-3 and TNF-α, were measured in the cortex, hippocampus, and dentate gyrus seven days post SAH. Compared to controls, there was no significant difference in either cortex or hippocampus cleaved caspase-3 between either the untreated animals or the animals treated with RTA 408. It was found to be significantly higher in the dentate gyrus in the untreated SAH group ($p <0.01$) and the SAH groups treated with either RTA 408 0.5 or 1 mg/kg/day ($p <0.05$) (Fig 3). Comparing the SAH only group with those treated with RTA 408 1.5 mg/kg/day, we found marked attenuation of cleaved caspase-3 induced by SAH in the dentate gyrus ($p <0.01$), restoring expression to control levels.

The results of our ELISA, revealed no marked differences in TNF-α in either the cortex or hippocampus between controls and untreated SAH rats or those treated with RTA 408. The untreated SAH rats and the SAH rats treated with RTA 408 0.5 mg/kg/day had significantly higher TNF-α expressions in the dentate gyrus group than the controls ($p <0.05$; Fig 4). In SAH rats treated with RTA 408 1.5 mg/kg/day, there was significant attenuation of SAH induced changes in TNF-α in dentate gyrus ($p <0.05$ vs. SAH only group).

## Discussion

This study examined the effect of RTA 408 on SAH in experimental rats. We found that RTA 408 attenuated SAH-induced vasospasm, achieving this effect by down-regulating NF-κB and iNOS after SAH. RTA 408 exerted its neuron-protective effect by first increasing the antioxidation protein Nrf2 and two of Nrf2's downstream proteins, HO-1 and NQO1. This was followed by decreases in SAH-induced protein-cleaved caspase-3 and TNF-α.

Although spontaneous SAH secondary to the rupture of intracranial aneurysms accounts for only 5% of strokes, morbidity and mortality associated with this disease remain high despite advances in the diagnosis and treatment of SAH aneurysms [23]. While SAH-induced CV contributes greatly to prognosis, it cannot alone fully explain the delayed neurological dysfunction that occurs from days 3–14 after an aneurysm rupture [3, 24]. Prior to this study, the pathophysiology underlying SAH-induced CV and secondary brain injury was unclear and adequate strategies for treating it were elusive. Increasing evidence has suggested that excess production of nitric oxide (NO) may contribute significantly to CV secondary to SAH [7]. Although SAH regulates the activity of all three isoforms of nitric oxide synthase (NOS), the inducible isoform iNOS accounts for most of the NO-mediated secondary damage after SAH [7]. Our study found SAH increased protein iNOS and it treatment with RTA 408 resulted in

significantly decreases both iNOS protein and CV, findings consistent with those of others showing suppression of iNOS expression can alleviate SAH-induced CV [25].

Recent and compelling findings have suggested that inflammation plays an important role in the development of SAH-induced CV, with various aspects of inflammation playing crucial roles its pathogenesis [26]. Specifically, the inflammation response activates iNOS, causing cellular damage through its generation of NO in an oxidative environment; iNOS, in turn, reacts with free radicals to form ROS, further exacerbating vascular inflammation. CV develops once SAH activates the inflammatory cascade [27]. The inflammation contributes to the upregulation of iNOS following SAH through its activation of one or more transcriptional pathways—such as NF-κB, hypoxia-inducible factor 1 (HIF-1), and the nuclear factor of activated T-cells (NFAT) [7]. NF-κB binds with its inhibitor, an enhancer in B-cell inhibitor (IκB) in the cytosol of cells. IκB can be degraded by several inflammatory signaling pathways which translocate it to the nucleus. Subsequently, the activated NF-κB binds to the promoter region for the iNOS gene and upregulates iNOS. Many pro-inflammatory agents have been reported to induce the expression of iNOS by their activation of NF-κB in various models of SAH [28–30]. Our findings as well as others suggest, because NF-kB has anti-spastic effects in SAH-induced CV, it can be targeted therapeutically to reduce SAH-induced increases in iNOS.

Clinically, some SAH patients have mild symptoms, no serious surgical complications, and radiographically no indication of vasospasm, though they have long-term psychological and neurobehavioral effects [31]. Around half of SAH survivors remain permanently disabled by cognitive dysfunction and unable to return to their previous jobs [32]. Because CV alone cannot explain all these changes in behavior and memory, we should consider the possible involvement of global ischemia related apoptosis [6]. Apoptosis has found in the vasculature with varying degrees of necrosis after SAH and has been implicated in the development of CV and smooth muscle cell proliferation in spastic arteries [33]. Furthermore, Zhou et al. have found that both pan caspase inhibitors and p53 inhibitors can attenuate post-SAH vasospasms, which suggests that apoptosis in the vasculature may be responsible for CV [34]. Thus, post-SAH apoptosis may have important implications CV as well as the long-term sequelae of SAH [6].

Various molecular mechanisms have been found to be involved in cellular apoptosis in secondary brain injury after SAH. These mechanisms include global ischemia [35], microcirculatory disturbance [36], and subarachnoid blood toxicity [37]. Apoptosis is activated via an intrinsic pathway mediated by the mitochondrial release of cytochrome C resulting in the activation of caspase-9 [37]. Caspase-9 activates downstream caspase-3, an apoptotic mediator well-known to lead to DNA damage and eventually apoptosis [35, 38]. Apoptosis-related proteins P53, cytochrome C, caspase-3, and caspase-8 have been found to be increased following SAH [38]. One study examining endovascular perforations in SAH rats have found apoptosis in cerebral endothelial cells as well as hippocampus and cortex [35]. SAH-induced apoptosis occurs through the activation of the NF-κB signaling pathway, followed by an increase in the levels of pro-inflammatory cytokines—such as iNOS, TNF-α, and the cytochrome c and caspase-3 released from mitochondria—which eventually leads to apoptosis. We found RTA 408 treatment to inhibit the NF-κB signaling pathway, thereby reducing apoptosis. It also indirectly inhibited NF-κB by activating the downstream Nrf2 signaling pathway, the anti-inflammatory cytokine HO-1, and the antioxidant NQO1, which also reduces apoptosis.

TNF-α is a pro-inflammatory cytokine involved in neuronal inflammation, apoptosis, and necrosis [39]. TNF-α has been found to be upregulated in both the cerebral cortex and hippocampus in an SAH animal model, and it may be responsible for SAH-induced apoptosis [39–41]. A variety of TNF-α inhibitors have been approved for the treatment of SAH-induced apoptosis. They include anti-tumor necrosis factor-alpha antibody [40], acetylcholine [39], and

ethyl pyruvate [41]. We found the treatment with RTA 408 reversed the increased expression of cleaved caspase-3 and TNF-α in the denate gyrus after SAH. We also combined RTA408 with Keap1 to increase the Nrf2 concentration in the cytoplasm and increase the level of downstream protein anti-inflammatory cytokines (such as HO-1). Meanwhile, HO-1 indirectly inhibits pro-inflammatory cytokines, such as TNF-α and iNOS. Additionally RTA408 either inhibits the NF-κB signaling pathway or indirectly reduces the decline of pro-inflammatory cytokines TNF-α and iNOS, thereby reducing apoptosis. Further studies may want to investigate the molecular mechanisms potentially underlying RTA 408's inhibition of SAH-induced apoptosis.

Oxidative stress not only causes CV but also causes delayed brain ischemia after SAH. Free haem acts as a catalyst for ROS formation and subsequent oxidative stress [42]. Following the conversion of OxyHb to MetHb, superoxide radicals are released and converted to hydroxyl radicals [43]. ROS causes post-SAH brain injury through the modification of lipids, carbohydrates, and nucleotides, where eventual cell death affects both neurons [44] and endothelial cells [45]. ROS produces vasoactive lipids that cause vasoconstriction and the free radical oxidation of bilirubin results in the formation of bilirubin oxidation products [46]. The accumulation of these products in the CSF has been associated with delayed cerebral ischemia and CV after SAH [47]. Oxidative stress has been linked to Nrf2, which has both antioxidant and anti-inflammatory effects and plays an important role in neuroprotection. RTA 408, an AIM, has been studied in relation to the treatment of various diseases, including neurodegenerative disorders, cancer, metabolic disorders, mitochondrial myopathies, and disorders with inflammatory and oxidative stress components [17–19, 48]. However, the clinical benefits of the Nrf2 activator in the treatment of SAH-induced complications remain uncertain.

RTA 408 achieves its effect by activating antioxidative transcription factor Nrf2 and inhibiting pro-inflammatory transcription factor NF-κB. Therefore, we examined its effect in on SAH in experimental rats and found that it decreased SAH-induced CV. SAH was found to downregulate Nrf2, HO-1, and NQO1 and upregulate NF-κB, iNOS, cleaved caspase-3, and TNF-α expression. RTA 408 reversed these SAH-induced changes. These findings indicate that it had anti-inflammation, antioxidation, and antiapoptosis effects (Fig 5). RTA 408 activates Nrf2 and inhibits NF-κB exerting both anti-inflammatory and anticarcinogenic effects. It increases Nrf2 levels by binding to the adaptor protein Keap1 and blocking its ability to promote Nrf2 degradation through Cul3-Rbx1-mediated ubiquitination and constitutive proteasomal degradation. Once Nrf2 is activated, it then moves to the cell nucleus where it binds to the ARE. This Nrf2–ARE binding regulates the levels of antioxidant genes and anti-inflammatory genes, NQO-1 and HO-1 for cellular antioxidant and anti-inflammatory defense. Nrf2 also inhibits the NF-κB pathway, reducing the level of pro-inflammatory cytokines, including TNF-α and iNOS, which in turn reduces inflammation and reduces apoptosis through its influence on the production of caspase-3 in the mitochondria In this study, RTA 408reduced the occurrence of CV and the subsequent occurrence of delayed brain ischemia in our SAH rats.

This study has several limitations. First, we employed a double injection model to mimic SAH, which usually induces mild secondary brain injury and neurological deficits compared to endovascular perforation model. Another limitation is that the basilar artery is located just below the magna cisterna and so we are now sure what the impact of distribution of blood is into the subarachnoid space. Experimentally, it may have been better to use an endovascular perforation model and tested MCA in our assessment of vasospasm after SAH. However, we still believe that the double injection model holds potential for the study of SAH -induced vasospasm and apoptosis. Another limitation is that we did little to evaluate secondary brain injury. Although cell death assay and TNF alpha in brain tissue were used to detect the brain

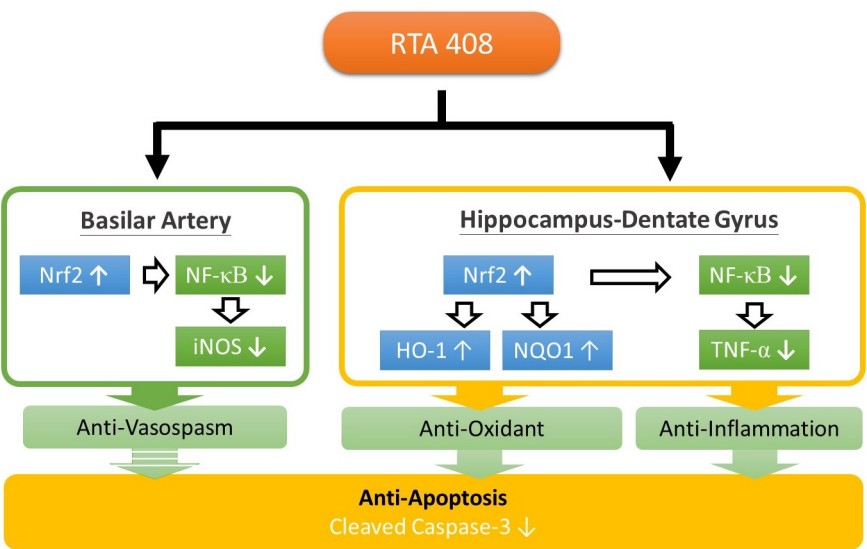

**Fig 5. The RTA 408 achieves its anti-inflammatory, anti-oxidant and anti-apoptotic effects via Nrf2 and NF-κB signaling pathway.**

cell apoptosis, more credible tools such as morphological, neuroimaging, molecular biological and neurobehavioral scores are needed to examine the secondary brain injury are need to monitor the secondary brain injury. Still another limitation is that we attempted to find a clear mechanism underlying the neuroprotective effects of RTA-408. Although we found changes in several Nrf2 downstream of protein levels, a clear mechanism was not elucidated. Therefore, additional intervention studies are still needed to verify the links between RTA408 and Nrf2 and the consequences of SAH. In addition, sham animals with injection of an artificial cerebral spinal fluid are required, at least in part, to evaluate the impact of anesthesia and surgery.

## Conclusion

This study found that RTA 408 attenuated SAH-induced vasospasm through its reversal of SAH-induced changes in Nrf2, NF-κB and iNOS. RTA 408 decreased both apoptosis-related protein-cleaved caspase-3 and the information cytokine TNF-α and thus had a neuron-protective effect, which could be further enhanced by Nrf2 and its downstream proteins HO-1 and NQO1. Additional studies are required to evaluate the antispastic and neuroprotective effects of RTA 408 as well as possible complications from its use. The mechanisms through which RTA 408 prevents SAH-induced CS and secondary brain injury also require further elucidation. We concluded that RTA 408, an Nrf2 activator, can potentially be used to manage SAH-induced CV and secondary brain injury, and it should be investigated further.

## Supporting information

**S1 Data.**
(PPTX)

## Author Contributions

**Conceptualization:** Tai-Hsin Tsai, Sheau-Fang Yang, Chih-Lung Lin.

**Data curation:** Szu-Huai Lin.

**Formal analysis:** Chieh-Hsin Wu, Chih-Lung Lin.

**Funding acquisition:** Chih-Lung Lin.

**Investigation:** Tai-Hsin Tsai, Szu-Huai Lin, Yi-Cheng Tsai, Chih-Lung Lin.

**Methodology:** Tai-Hsin Tsai, Szu-Huai Lin, Yi-Cheng Tsai, Chih-Lung Lin.

**Project administration:** Tai-Hsin Tsai, Chieh-Hsin Wu, Sheau-Fang Yang, Chih-Lung Lin.

**Resources:** Tai-Hsin Tsai.

**Validation:** Tai-Hsin Tsai, Chieh-Hsin Wu, Chih-Lung Lin.

**Writing – original draft:** Tai-Hsin Tsai, Yi-Cheng Tsai, Sheau-Fang Yang, Chih-Lung Lin.

**Writing – review & editing:** Tai-Hsin Tsai, Yi-Cheng Tsai, Sheau-Fang Yang, Chih-Lung Lin.

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
