## [Decision Letter · Decision Letter 0]

29 Jul 2020

PONE-D-20-19417

Mechanisms and therapeutic implications of RTA 408, an activator of Nrf2, in subarachnoid hemorrhage–induced delayed cerebral vasospasm and secondary brain injury

PLOS ONE

Dear Dr. Lin,

Thank you for submitting your manuscript to PLOS ONE. After careful consideration, we feel that it has merit but does not fully meet PLOS ONE’s publication criteria as it currently stands. Therefore, we invite you to submit a revised version of the manuscript that addresses the points raised during the review process.

We look forward to receiving your revised manuscript.

Kind regards,

Jinglu Ai, M.D., Ph.D.

Academic Editor

PLOS ONE

Journal Requirements:

2. At this time, we request that you please report additional details in your Methods section regarding animal care, as per our editorial guidelines:

(1) Please state the source of the rats used in the study  

(2) Please provide details of animal welfare (e.g., shelter, food, water, environmental enrichment)

(3) Please include the method of euthanasia and clarify whether the rats were anaesthetised before they were euthanized by perfusion and fixatiion. Please provide the reagents used in the method of euthanasia.  

(4) Please describe the post-operative care received by the animals, including the frequency of monitoring and the specific clinical, physiological and behavioural criteria used to assess animal health and well-being.

Thank you for your attention to these requests.

3. In the Methods section, please provide the product number and any lot numbers of the primary antibodies used in the Western blot analysis for your study.

4. In the Methods section, please provide the source, product number and any lot numbers of the RTA-408 used in the animal experiments for your study.

5. To comply with PLOS ONE submission guidelines, in your Methods section, please provide additional information regarding your statistical analyses, including the specific name and version of the software used. For more information on PLOS ONE's expectations for statistical reporting, please see https://journals.plos.org/plosone/s/submission-guidelines.#loc-statistical-reporting.

6. We noticed minor instances of text overlap with the following previous publication(s), which need to be addressed:

(1) https://www.sciencedirect.com/science/article/abs/pii/S1878331712001313?via%3Dihub

(2) https://thejns.org/view/journals/j-neurosurg/110/3/article-p457.xml

The text that needs to be addressed involves the (1) Introduction and (2) Results section (pages 8-9).

In your revision please ensure you cite all your sources (including your own works), and quote or rephrase any duplicated text outside the methods section. Further consideration is dependent on these concerns being addressed.

7. We note that you have indicated that data from this study are available upon request. PLOS only allows data to be available upon request if there are legal or ethical restrictions on sharing data publicly. For information on unacceptable data access restrictions, please see http://journals.plos.org/plosone/s/data-availability#loc-unacceptable-data-access-restrictions.

8. PLOS ONE now requires that authors provide the original uncropped and unadjusted images underlying all blot or gel results reported in a submission’s figures or Supporting Information files. This policy and the journal’s other requirements for blot/gel reporting and figure preparation are described in detail at https://journals.plos.org/plosone/s/figures#loc-blot-and-gel-reporting-requirements and https://journals.plos.org/plosone/s/figures#loc-preparing-figures-from-image-files. When you submit your revised manuscript, please ensure that your figures adhere fully to these guidelines and provide the original underlying images for all blot or gel data reported in your submission. See the following link for instructions on providing the original image data: https://journals.plos.org/plosone/s/figures#loc-original-images-for-blots-and-gels.

9. PLOS requires an ORCID iD for the corresponding author in Editorial Manager on papers submitted after December 6th, 2016. Please ensure that you have an ORCID iD and that it is validated in Editorial Manager. To do this, go to ‘Update my Information’ (in the upper left-hand corner of the main menu), and click on the Fetch/Validate link next to the ORCID field. This will take you to the ORCID site and allow you to create a new iD or authenticate a pre-existing iD in Editorial Manager. Please see the following video for instructions on linking an ORCID iD to your Editorial Manager account: https://www.youtube.com/watch?v=_xcclfuvtxQ

Reviewers' comments:

Reviewer's Responses to Questions

**Comments to the Author**

1. Is the manuscript technically sound, and do the data support the conclusions?

Reviewer #1: Yes

2. Has the statistical analysis been performed appropriately and rigorously? 

Reviewer #1: Yes

3. Have the authors made all data underlying the findings in their manuscript fully available?

Reviewer #1: Yes

4. Is the manuscript presented in an intelligible fashion and written in standard English?

Reviewer #1: Yes

5. Review Comments to the Author

Reviewer #1: The authors have examined the potential efficacy of RTA 408 in attenuating cerebral vasospasm and acute brain injury following SAH in a rodent model.

There are some minor grammatical errors that should be corrected.

How were the dosages of RTA 408 selected?

Did the authors examine Nrf2 nuclear translocation?

A limitations section should be provided.

6. PLOS authors have the option to publish the peer review history of their article (what does this mean?). If published, this will include your full peer review and any attached files.

Reviewer #1: No

---

## [Author Response · Author response to Decision Letter 0]

8 Sep 2020

Dear reviewers:

Thank you for the opportunity to revise our paper on ‘Mechanisms and therapeutic implications of RTA 408, an activator of Nrf2, in subarachnoid hemorrhage–induced delayed cerebral vasospasm and secondary brain injury.’ I have included the reviewer comments immediately and responded to them individually, indicating exactly how we addressed each concern or problem and describing the changes we have made. The revisions have been approved by all authors. The changes are marked in red in the paper as you requested, and the revised manuscript is attached to this email message.

Reply to reviewer's comment

Reviewer 1

Answer:

The manuscript has meet PLOS ONE's style requirements as the PLOS ONE'S style templates.

2. At this time, we request that you please report additional details in your Methods section regarding animal care, as per our editorial guidelines

Answer: 

We have been reported the additional details in Methods section regarding animal care such as the source of the rats, animal welfare, the method of euthanasia and postoperative care. Page 4-5

3. In the Methods section, please provide the product number and any lot numbers of the primary antibodies used in the Western blot analysis for your study.

Answer: 

We have been provided the product numbers of the primary antibodies used in the Western blot analysis in this manuscript. Page 6

4. In the Methods section, please provide the source, product number and any lot numbers of the RTA-408 used in the animal experiments for your study.

Answer:

We have been provided the source, product number of the RTA 408 used in the animal experiments. Page 5

5. To comply with PLOS ONE submission guidelines, in your Methods section, please provide additional information regarding your statistical analyses, including the specific name and version of the software used. 

Answer:

We have been provided the specific name and version of the software regarding the specific name and version of the software in the Methods section. Page 6

6. We noticed minor instances of text overlap with the following previous publication(s), which need to be addressed involves the (1) Introduction and (2) Results section (pages 8-9).

Answer:

We also noticed the text overlap with the previous publications. Although the previous publications are our own works, we have been rephrased and quoted the duplicate text to avoid plagiarism or self-plagiarism. Page2-3, 7-8

7. We note that you have indicated that data from this study are available upon request. PLOS only allows data to be available upon request if there are legal or ethical restrictions on sharing data publicly. 

Answer:

Data from this study are available upon request and there are no restrictions on sharing data publicly. We will upload the data set necessary to replicate our study findings as Supporting Information files. 

8. PLOS ONE now requires that authors provide the original uncropped and unadjusted images underlying all blot or gel results reported in a submission’s figures or Supporting Information files. 

Answer:

We will provide the original uncropped and unadjusted images underlying all blot or gel results reported in a submission’s figures.

9. PLOS requires an ORCID iD for the corresponding author in Editorial Manager on papers submitted after December 6th, 2016. 

Answer:

We have an ORCID iD for the corresponding author and that it is validated in Editorial Manager. 

Reviewer 2: 

1. There are some minor grammatical errors that should be corrected.

Answer:

The grammatical errors of the article have been edited.

2. How were the dosages of RTA 408 selected?

Answer:

In this experiment, Group 3-5 received experimental SAH with RTA 408 treatment (Intraperitoneal injection) 30 min after the first induction of SAH for 7 days following the first hemorrhage. The dosages of RTA 408 are 0.5 mg/kg/day, 1 mg/kg/day and 1.5 mg/kg/day. The dosages of RTA 408 were selected according to the previous experiments.Peng Han, et al. RTA-408 Protects Kidney from Ischemia-Reperfusion Injury in Mice via Activating Nrf2 and Downstream GSH Biosynthesis Gene. Oxid Med Cell Longev. 24 December 2017. Page 4-5

3. Did the authors examine Nrf2 nuclear translocation?

Answer：

The authors do not examine Nrf2 nuclear translocation, because this experimental design is for the study of downstream protein changes in the Nrf2 pathway caused by RTA 408 treatment. Despite the changes of several Nrf2 downstream of protein levels after RTA-408 treatment, the authors did not directly sought out a clear mechanism for the neuroprotective effects of RTA-408. Therefore, additional intervention studies including Nrf2 nuclear translocation are still needed to verify the links between RTA408 and Nrf2 and the consequences of SAH. This problem was addressed in the limitation section.

4. A limitations section should be provided.

Answer:

A limitation section has been provided. Page 12

We hope the revised manuscript will better suit the PLOS ONE but are happy to consider further revisions, and we thank you for your continued interest in our research.

Sincerely,

Tai-Hsin Tsai, MD

Department of Neurosurgery

Kaohsiung Medical University Hospital

---

## [Editor Report · Decision Letter 1]

11 Sep 2020

PONE-D-20-19417R1

Mechanisms and therapeutic implications of RTA 408, an activator of Nrf2, in subarachnoid hemorrhage–induced delayed cerebral vasospasm and secondary brain injury

PLOS ONE

Dear Dr. Lin,

Thank you for submitting your manuscript to PLOS ONE. After careful consideration, we feel that it has merit but does not fully meet PLOS ONE’s publication criteria as it currently stands. Therefore, we invite you to submit a revised version of the manuscript that addresses the points raised during the review process.

Additional Editor Comments (if provided):

Thank you for the revision. Before I accept your revised manuscript. I would like you to revise all your figures to make them look more professional.

E.g., Figure 1, the y-axis label is not centered to the axis, and the labeling of the three doses of the drug is way too long. You could simply label them as 0.5 mg, 1 mg and 1.5 mg, and on top of them use a bracket to include all three, and label the bracket with text, SAH+RTA408. This is the same for the chart labeling as well, and for all the figures. After the modification, and save some spaces, you may want to increase the font size for all the labeling on the charts.

In the PDF file, the figures, especially the charts in the figures are of very bad quality, some are even not readable, especially the last figure with 12 charts. I am not sure whether the journal office has a better version of the figures. It is not publishable as presented in the current PDF version.

Thank you,

Jinglu Ai

Please submit your revised manuscript by Sept 21st. If you will need more time than this to complete your revisions, please reply to this message or contact the journal office at plosone@plos.org. Please include the following items when submitting your revised manuscript:

We look forward to receiving your revised manuscript.

Kind regards,

Jinglu Ai, M.D., Ph.D.

Academic Editor

PLOS ONE

---

## [Author Response · Author response to Decision Letter 1]

17 Sep 2020

Reply to reviewer's comment

Additional Editor Comments:

1.Please revise all your figures to make them look more professional and I have uploaded a copy o Figure 5.

Answer : 

I have revised all figures to make them look more professional.

Reviewer 1

Answer:

The manuscript has meet PLOS ONE's style requirements as the PLOS ONE'S style templates.

2. At this time, we request that you please report additional details in your Methods section regarding animal care, as per our editorial guidelines

Answer: 

We have been reported the additional details in Methods section regarding animal care such as the source of the rats, animal welfare, the method of euthanasia and postoperative care.

3. In the Methods section, please provide the product number and any lot numbers of the primary antibodies used in the Western blot analysis for your study.

Answer: 

We have been provided the product numbers of the primary antibodies used in the Western blot analysis in this manuscript.

4. In the Methods section, please provide the source, product number and any lot numbers of the RTA-408 used in the animal experiments for your study.

Answer:

We have been provided the source, product number of the RTA 408 used in the animal experiments.

5. To comply with PLOS ONE submission guidelines, in your Methods section, please provide additional information regarding your statistical analyses, including the specific name and version of the software used. 

Answer:

We have been provided the specific name and version of the software regarding the specific name and version of the software in the Methods section.

6. We noticed minor instances of text overlap with the following previous publication(s), which need to be addressed involves the (1) Introduction and (2) Results section (pages 8-9).

Answer:

We also noticed the text overlap with the previous publications. Although the previous publications are our own works, we have been rephrased and quoted the duplicate text to avoid plagiarism or self-plagiarism.

7. We note that you have indicated that data from this study are available upon request. PLOS only allows data to be available upon request if there are legal or ethical restrictions on sharing data publicly. 

Answer:

Data from this study are available upon request and there are no restrictions on sharing data publicly. We will upload the data set necessary to replicate our study findings as Supporting Information files. 

8. PLOS ONE now requires that authors provide the original uncropped and unadjusted images underlying all blot or gel results reported in a submission’s figures or Supporting Information files. 

Answer:

We will provide the original uncropped and unadjusted images underlying all blot or gel results reported in a submission’s figures.

9. PLOS requires an ORCID iD for the corresponding author in Editorial Manager on papers submitted after December 6th, 2016. 

Answer:

We have an ORCID iD for the corresponding author and that it is validated in Editorial Manager. 

Reviewer 2: 

1. There are some minor grammatical errors that should be corrected.

Answer:

The grammatical errors of the article have been edited and the editing certificate is attached.

2. How were the dosages of RTA 408 selected?

Answer:

In this experiment, Group 3-5 received experimental SAH with RTA 408 treatment (Intraperitoneal injection) 30 min after the first induction of SAH for 7 days following the first hemorrhage. The dosages of RTA 408 are 0.5 mg/kg/day, 1 mg/kg/day and 1.5 mg/kg/day. The dosages of RTA 408 were selected according to the previous experiments. Peng Han, et al. RTA-408 Protects Kidney from Ischemia-Reperfusion Injury in Mice via Activating Nrf2 and Downstream GSH Biosynthesis Gene. Oxid Med Cell Longev. 24 December 2017.

3. Did the authors examine Nrf2 nuclear translocation?

Answer：

The authors do not examine Nrf2 nuclear translocation, because this experimental design is for the study of downstream protein changes in the Nrf2 pathway caused by RTA 408 treatment. Despite the changes of several Nrf2 downstream of protein levels after RTA-408 treatment, the authors did not directly sought out a clear mechanism for the neuroprotective effects of RTA-408. Therefore, additional intervention studies including Nrf2 nuclear translocation are still needed to verify the links between RTA408 and Nrf2 and the consequences of SAH. This problem was addressed in the limitation section.

4. A limitations section should be provided.

Answer:

A limitation section has been provided.

---

## [Editor Report · Decision Letter 2]

21 Sep 2020

Mechanisms and therapeutic implications of RTA 408, an activator of Nrf2, in subarachnoid hemorrhage–induced delayed cerebral vasospasm and secondary brain injury

PONE-D-20-19417R2

Dear Dr. Lin,

We’re pleased to inform you that your manuscript has been judged scientifically suitable for publication and will be formally accepted for publication once it meets all outstanding technical requirements.

Kind regards,

Jinglu Ai, M.D., Ph.D.

Academic Editor

PLOS ONE
---

## [Editor Report · Acceptance letter]

25 Sep 2020

PONE-D-20-19417R2 

Mechanisms and therapeutic implications of RTA 408, an activator of Nrf2, in subarachnoid hemorrhage–induced delayed cerebral vasospasm and secondary brain injury 

Dear Dr. Lin:

I'm pleased to inform you that your manuscript has been deemed suitable for publication in PLOS ONE. Congratulations! Your manuscript is now with our production department. 

Kind regards, 

on behalf of

Dr. Jinglu Ai 

Academic Editor

PLOS ONE